# Seismic and Newtonian Noise in the GW Detectors

**Lucia Trozzo** [1,*,†] and **Francesca Badaracco** [2,*,†]

1 Istituto Nazionale di Fisica Nucleare, Sezione di Napoli, Strada Comunale Cinthia, 80126 Napoli, Italy

2 Centre for Cosmology, Particle Physics and Phenomenology, Université Catholique de Louvain,
  B-1348 Louvain-La-Neuve, Belgium

\* Correspondence: lucia.trozzo@na.infn.it (L.T.); francesca.badaracco@uclouvain.be (F.B.)

† These authors contributed equally to this work.

**Abstract:** Gravitational wave detectors aim to measure relative length variations of the order of $\Delta L/L \simeq 10^{-21}$, or less. Thus, any mechanism that is able to reproduce such a tiny variation can, in principle, threaten the sensitivity of these instruments, representing a source of noise. There are many examples of such noise, and seismic and Newtonian noise are among these and will be the subject of this review. Seismic noise is generated by the incessant ground vibration that characterizes Earth. Newtonian noise is instead produced by the tiny fluctuations of the Earth's gravitational field. These fluctuations are generated by variations of air and soil density near the detector test masses. Soil density variations are produced by the same seismic waves comprising seismic noise. Thus, it makes sense to address these two sources of noise in the same review. An overview of seismic and Newtonian noise is presented, together with a review of the strategies adopted to mitigate them.

**Keywords:** seismic noise; Newtonian noise; seismic isolation system; noise subtraction



## 1. Introduction

Current gravitational wave (GW) detectors are sensitive to signals in a frequency band that ranges between 10 Hz and 10 kHz. Their sensitivity can be affected by several dominant sources of noise, such as ground motion [1], local terrestrial gravity [2], magnetic noise [3], thermal [4] and quantum noise [5].

These noises cause a phase variation of the laser light detected at the interferometer output port in the same way that a passing GW would. If we want to increase the detectors' sensitivity to GWs, it is necessary to reduce these noises as much as possible.

Seismic noise couples with GW detectors via vibrations transmitted through the suspension system and other isolation systems. On the other hand, seismic noise can also directly couple with the detector through what is known in GW physics as Newtonian noise (NN). It is well known from Newtonian physics that a variation in mass density leads to fluctuations in the surrounding gravitational field. Therefore, a passing seismic wave that causes density variations will also produce gravity fluctuations, and thus NN.

In the following sections, we will focus on seismic and Newtonian noise. In Section 2, the origin of seismic noise will be briefly discussed for surface and underground environments. In Section 3, the technologies adopted to mitigate seismic noise will be discussed. In Section 4, a brief introduction of NN will be given, together with a description of atmospheric NN (ANN) and seismic NN (SNN). In Section 5, other sources contributing to NN will be shortly addressed. Finally, the noise subtraction techniques adopted to mitigate NN will be discussed in Section 6.

## 2. An Introduction to Seismic Noise in GW Detectors

Ground-based GW detector performance is affected by so-called seismic noise. This noise affects detector sensitivity in the range 0.1–10 Hz, also undermining the possibility of operating with a high duty cycle.

What is seismic noise? In geophysics, seismic noise is used to identify the persistent and variable ground vibration that can be detected everywhere on the Earth and that produces a characteristic spectrum.

This kind of vibration is generated by **seismic waves** travelling through the Earth's layers at different speeds and produced by phenomena such as wind, ocean waves, earthquakes, anthropogenic sources, etc. Seismic wave speeds can change depending on the density and the elasticity of the crossed medium, as well as on their depth. Moving from the Earth's crust to the deep mantle, the speed increases from a few m/s up to 13 km/s [6,7].

Seismic waves can be classified as body waves and surface waves.

- **Body waves** include all those waves travelling through the Earth. The density and stiffness of the crossed material depend on temperature, chemical composition, and material phase. For this reason, body waves show different velocities with increasing depth of propagation. They are classified into primary waves (P-waves) and secondary waves (S-waves). P-waves cause a compression/decompression displacement along their propagation direction, whereas S-waves produce a shear displacement perpendicular to their propagation direction. In an earthquake event, S-waves are slower than P-waves (with a typical speed value of about 60% of that of P-waves). S-waves can only travel through solid materials. Indeed, fluids (liquids and gases) do not support shear stresses (see Figure 1). P-waves generate density variations in the medium, so they produce gravity fluctuations. S-waves, being shear waves, can instead produce density variations only in presence of some discontinuity (see Section 4.2).

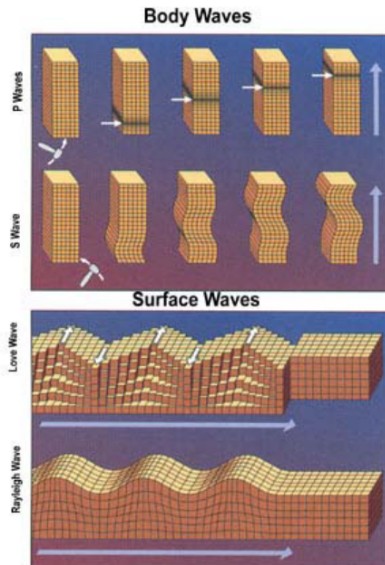

**Figure 1.** Body and surface waves [8].

- **Surface waves** travel on the Earth's surface and their amplitude decreases exponentially as a function of their depth from the surface. Their speed is slower compared to the speed of body waves (P and S) and their amplitudes can reach several cm in an earthquake event. Mathematically, they arise from the interaction of body waves with a medium discontinuity. They can be distinguished into **Rayleigh waves** (or ground rolls) and **Love waves**. Rayleigh waves produce both longitudinal and transverse motion of surface particles, generating a retrograde vertical ellipse motion in the plane normal to the surface and containing the wave's propagation direction. In a homogeneous and isotropic half-space, Rayleigh waves have a slightly lower velocity than S-waves ($v_R \simeq 0.9\ v_S$) and are non-dispersive (the velocity does not depend on the frequency). Instead, if the half-space is composed of homogeneous and isotropic

layers, the waves become dispersive. The dispersion model has an important impact on the gravity fluctuations generated by Rayleigh waves (see Section 4.2).

Love wave polarization is perpendicular to the propagation direction and parallel to the surface. Moreover, these waves have larger amplitudes and speed compared to Rayleigh waves (see Figure 1). Contrary to Rayleigh waves, they cannot propagate in homogeneous and isotropic half-space, but they arise in layered mediums, showing a dispersive model. Being surface shear waves, they do not produce density variations; thus, they are not a source of gravity fluctuations.

### 2.1. Seismic Noise: NLNM and NHNM Models

Seismic noise is detected and measured with broad-band seismographs everywhere on the Earth. The spectral density associated with this noise is modeled approximately by

$$S(f) \simeq \frac{\alpha}{f^2} \qquad (1)$$

where $\alpha$ depends on the site. For example, the value of $\alpha$ in the Italian site (Cascina, PI) hosting the Virgo detector is about $10^{-7}$ m $(\mathrm{Hz})^{\frac{3}{2}}$.

With the purpose of monitoring variations in the seismic noise spectral density, in 1993, J. Peterson collected and analysed data from 75 seismic stations around the world. He developed two seismic noise models: the New Low Noise Model (NLNM) and the New High Noise Model (NHNM) [9]. These models, now commonly used as a reference in the scientific community, represent the lowest and the highest seismic background spectra that it is possible to find on the Earth. They were obtained by fitting with straight lines the lower and the upper envelopes of all the spectra measured.

According to Peterson's results (Figure 2), it is possible to split the seismic spectrum into different regions. The region in the 0.05–1 Hz band is dominated by so-called **microseismic noise**, generated every time atmospheric phenomena such as typhoons, storms, and climatic variations occur [10,11]. The most energetic seismic waves that comprise microseisms are Rayleigh waves, but Love and body waves can also contribute.

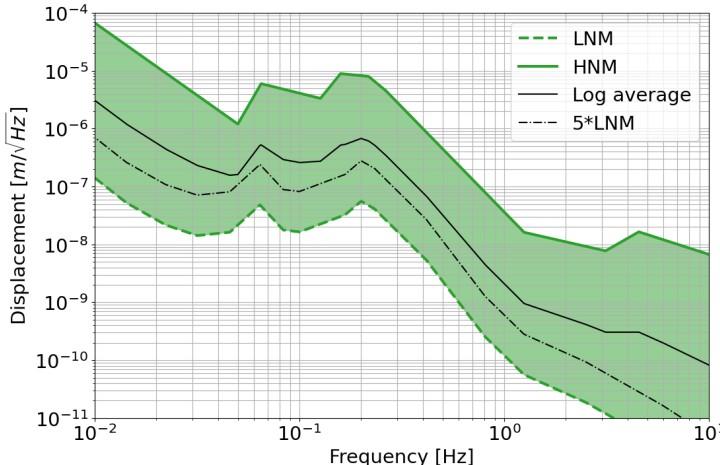

**Figure 2.** Square root of the power spectral density of the seismic displacement of Peterson's New Low and High Noise Models (NLNM and NHNM), the average of their logarithmic values and 5 times the NLNM .

Microseism signals are non-impulsive, and their amplitudes show strong seasonal modulation, with maxima during winter seasons (when oceans are stormier) and minima during summers. Looking at Figure 2, microseismic noise displays two predominant peaks in the region of 0.05–1 Hz. The weaker peak between 0.05–0.1 Hz is called the "primary peak" and it is explained by the effect of surface gravity waves in shallow waters. It shares the same spectral content as the ocean waves. Its source is associated with the energy

transfer of ocean waves breaking on the coast. The stronger peak between 0.1–1 Hz is called the "secondary peak" and it is generated by interactions between waves of the same frequency travelling in opposite directions [12]. It is characterized by twice the frequency of the ocean waves. This peak generally shows a higher amplitude than the primary one.

At frequencies $f < 0.05$ Hz, phenomena such as atmospheric gravity fluctuations and **tidal effects** (generated by the Sun–Moon gravitational attraction), start to manifest themselves in the seismic spectrum, whereas all the vibrations included in the range 1–10 Hz are classified as **anthropogenic noise**, i.e., all the noise produced by human activities such as industrial processes, vehicles, agricultural machinery, wind turbines, etc.

## 2.2. Seismic Noise in Underground Sites

Seismic classification between low-frequency ($f < 1$ Hz) and high-frequency ($f > 1$ Hz) noise is a way to distinguish natural phenomena from human ones. The heterogeneity of the Earth's crust influences the microseismic spectrum, decreasing or increasing the spectral amplitudes at a specific frequency $f$. At high frequencies, instead, the amplitude of the seismic spectrum is mainly associated with industrial and transportation activities, but also with drastic weather changes. Seismic noise in underground locations (as well as surface ones) depends on the proximity to the coast, urban areas, and on their geological history.

In underground sites, the atmospheric perturbation influence is minimal. The environmental conditions are more stable; thus, seismic noise is attenuated with respect to the surface. Indeed, the energy content of seismic noise is mainly carried by surface waves, which decay exponentially with increasing depth.

With these considerations in mind, the spectral density of seismic noise in underground locations (depth $> 100$ m) can be modelled approximately by

$$\frac{10^{-9}}{f^2} \, \mathrm{m} \, \mathrm{Hz}^{\frac{3}{2}} \tag{2}$$

Comparing Equations (1) and (2), it is evident that seismic noise in underground locations is about 100 times smaller than in surface ones. This can be seen when looking at the spectral noise profiles measured at 1 Hz in some mines: at Kamioka (the experimental site of KAGRA) and Sos-Enattos (Sardinia, Italy) we have $S(f) \simeq \frac{10^{-9} \cdot \mathrm{m}}{\sqrt{\mathrm{Hz}}}$, whereas at Homestake (USA) we have $S(f) \simeq \frac{1.5 \cdot \mathrm{m}}{\sqrt{\mathrm{Hz}}}$. These values are close to the NLNM curve: $S(f) \simeq \frac{10^{-10} \cdot \mathrm{m}}{\sqrt{\mathrm{Hz}}}$ (see Figures 3 and 4).

These values—their geological stability and their distance from industrial activities of underground locations—are the the main reasons why these are considered good candidate sites for the construction of the Einstein Telescope (ET), a third-generation GW detector.

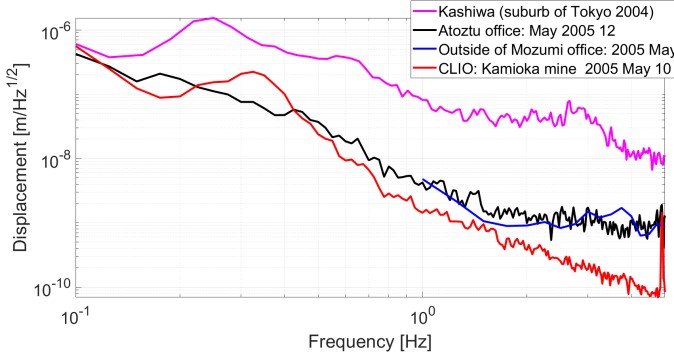

**Figure 3.** This figure shows the typical spectral density of the seismic noise displacement measured in an underground location at Kamioka (red curve) and compared to those measured at the surface (Tokyo areas). This plot was made using the data shown in Figure 4 in [13].

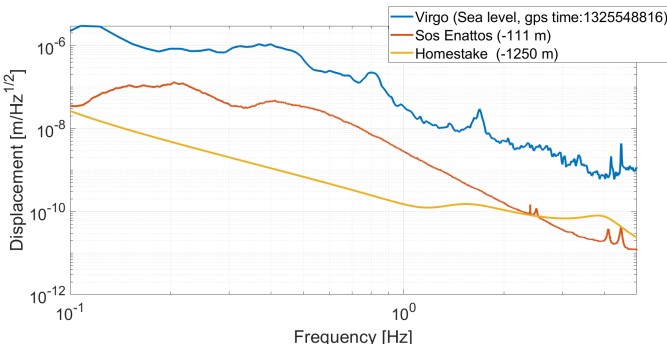

**Figure 4.** This plot shows the spectral density of the seismic noise displacement measured at the Sos-Enattos mine, compared to those measured near the Central building of Virgo (Pisa, Italy) and at the former mine of Homestake (Lead, SD -USA). The Homestake seismic profile was reconstructed using the data shown in the bottom right of Figure 6.10 in [14].

### 3. Mechanical Attenuators

Suspending optical components represents a crucial task for the construction of GW interferometers. In fact, the observation of GWs depends on the presence of free-falling (test) masses in the experimental apparatus. For this reason, the test masses must be well isolated (i.e., suspended) from ground vibrations in the frequency band relevant to scientific observations [15].

As discussed in the previous section, seismic noise in the frequency range 0.1–10 Hz represents one of the main limitations to the target sensitivity of GW detectors. Even in underground environments, where the seismic noise is lower than at the surface, ground displacement is eight orders of magnitude larger than that induced by a GW (see Figure 5).

To isolate and suppress the seismic noise transmitted from the ground to the test masses, a GW detector requires complex mechanical suspensions that are essentially low-pass filters and that allow one to consider the test masses as free-falling bodies in the horizontal direction (this is true starting from a few Hz).

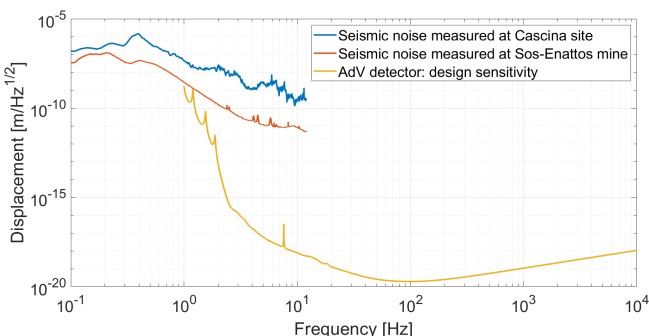

**Figure 5.** Ground displacement measured at the Cascina site (blue curve) and at the Sos-Enattos mine (red curve) compared to the design sensitivity of the Advanced Virgo detector (AdV). At 10 Hz, the ground displacement is about eight orders of magnitude higher than the sensitivity of the GW detector.

A free-falling mass can be fairly approximated by suspending it through a simple pendulum: this solution allows one to filter out the seismic noise that otherwise would affect the mass.

In order to describe the mechanical response of a simple pendulum to a generic disturbance, it is important to introduce the concept of a mechanical transfer function. In the frequency domain, the transfer function of a system is a linear operator that relates the system's input ($x_i(\omega)$) to its output ($x_o(\omega)$):

$$H(\omega) = \frac{x_o(\omega)}{x_i(\omega)} \tag{3}$$

The transfer function of a simple pendulum subject to internal friction can be calculated starting from its equation of motion:

$$-M\omega^2 x_o = -k(1 + i\phi)(x_o - x_i) \tag{4}$$

where $x_i$ and $x_o$ respectively represent the (input) displacement caused by the ground vibrations of the point from which the pendulum is suspended and the (output) displacement of the suspended mass. $M$ is the suspended body mass, $k$ is the oscillator stiffness, and $\phi$ is the dissipation factor of the system. By taking the ratio between $x_o$ and $x_i$ and using some algebra, it is possible to express the mechanical transfer function magnitude as

$$|H(\omega)| = \frac{\sqrt{1 + \phi^2}}{\sqrt{(1 - \frac{\omega^2}{\omega_0^2})^2 + \phi^2}} \tag{5}$$

where $\omega_0$ is the resonant mode of the system, defined as $\sqrt{\frac{k}{M}}$. The quality factor $Q$ of such a system is defined as $Q = \frac{1}{\phi}$.

With reference to Figure 6 (blue curve) and considering a harmonic oscillator with resonance mode $\omega_0$ and quality factor $Q$, we observe that:

- At frequency $\omega \ll \omega_0$, the transfer function is equal to 1 and the force is totally transmitted from the input to the output;
- When $\omega = \omega_0$, the applied force to the input determines a continuous transfer of energy from the input to the output and an amplification of the system motion with a quality factor $Q$;
- At frequency $\omega \gg \omega_0$, the system behaves as a second-order low-pass mechanical filter.

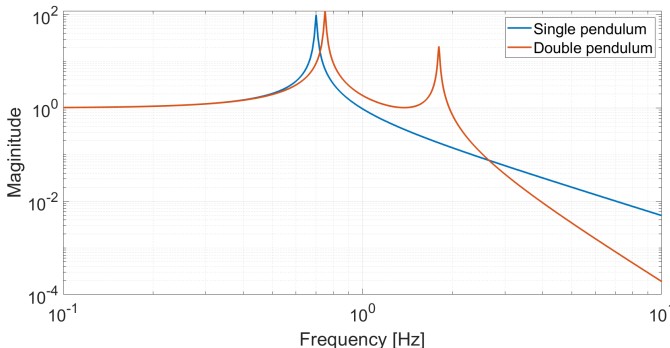

**Figure 6.** Mechanical transfer functions of a single (blue curve) and a double pendulum (red curve) having the same length.

Increasing the number of stages in a mechanical structure represents a good approach to improving the total attenuation factor of seismic noise at the level of the test masses. In such a way, the magnitude of the transfer function of a cascade of N harmonic oscillators (N stages) can be written as

$$|H(\omega)| = \prod_{i=1}^{N} \frac{\sqrt{1 + \phi_i^2}}{\sqrt{(1 - \frac{\omega^2}{\omega_{0i}^2})^2 + \phi_i^2}} \tag{6}$$

Above the resonances of the chain, such a system behaves as a low-pass filter of order $2^N$ that is able to inhibit the transmission of the seismic noise to the suspended body.

### 3.1. Seismic Isolation Systems

The seismic isolation systems adopted in current GW detectors are based on the idea of suspending a chain of harmonic oscillators in a cascade to filter out the transmitted noise at the test mass level in all its degrees of freedom.

To fulfil the detector requirements, during the last twenty years, a great effort has been made to develop these kind of systems. A few important guidelines were followed: the normal modes of the pendulum mechanical structures must be confined in the low frequency region (below 5 Hz); the pre-isolation stage is used as a mechanical support for the suspension point of the chain, allowing to move it by applying small forces; and finally, the active isolation platform should provide a good level of seismic isolation.

In the following, we will give a brief description of the seismic isolation system used in the LIGO, KAGRA, and Virgo systems.

### 3.1.1. LIGO Seismic Isolation System

The seismic isolation system adopted by LIGO uses a combination of active and passive stages [16–18]. A conceptual and a CAD model of the **BSC (Basic Symmetric Chamber)**, which houses the test mass and the transfer function of the quadruple pendulum, are shown in Figure 7. The BSC chamber, a 4.5 m tall suspension, consists of three cascade systems. The first system, the active Hydraulic External Pre-Isolator (The HEPI), provides the first isolation stage. The second system, the Internal Seismic Isolation platform (BSC-ISI), provides two stages of isolation. The test mass is suspended by 480 micron-fused silica fibres to a quadruple pendulum. This system shows resonance frequencies ranging from 0.45 to 4 Hz; thus, the isolation from ground vibrations is provided above these frequencies. Finally, vertical attenuation is provided by maraging steel blades installed in the first three stages of the quadruple pendulum.

A remarkable attenuation in the microseismic region is achieved through the use of active controls and noise subtraction techniques [19–22].

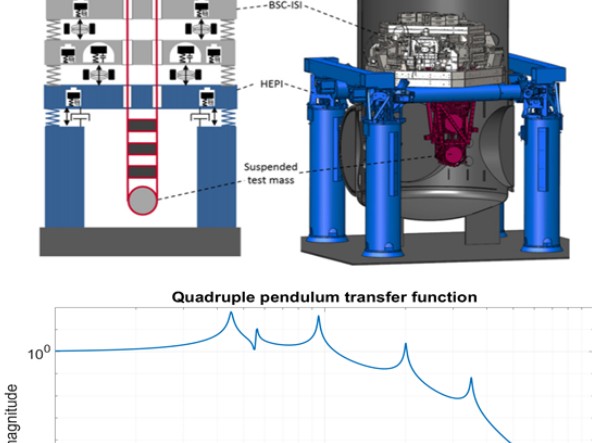

**Figure 7.** Schematic model (**a**) and technical drawing (**b**) of the BSC chamber supporting the LIGO test masses [17]. For completeness, panel (**c**) shows the quadruple pendulum mechanical transfer function from the ground to the test mass along the longitudinal degree of freedom. This transfer function was reconstructed following Figure 5 in [18].

### 3.1.2. KAGRA Seismic Isolation System

KAGRA differs from LIGO and Virgo. Indeed, it is the first GW detector built completely underground with cryogenic technologies [23]. Its suspension concept takes inspiration from the Virgo one and it is called the **Type A system**. With reference to Figure 8, it consists of a multi-stage pendulum that is 13.5 m tall. It is based on the second floor of an underground mine, supported by three inverted pendulum legs (1 m). The payload, comprising four stages, is installed inside a cryostat and cooled down to 20 K. The vertical attenuation is provided by triangular maraging blades installed in all the filters present in the chain, which are called Geometrical Anti Spring (GAS) filters.

The attenuation of the suspension is passive but active controls damp the structural modes in the 0.07–3 Hz band.

### 3.1.3. Virgo Seismic Isolation System

Virgo test masses are suspended using the so-called **Super-Attenuator (SA)**, which is composed of a multi-stage seismic isolation system [24,25]. Referring to Figure 9, it consists of a pre-isolator, which is a soft inverted pendulum (30 mHz), from which is suspended a 9.2 m long multistage chain connected by means of a single wire at the center of mass of each stage. This chain comprises a first filter at the top of the inverted pendulum (filter 0), four standard filters (the pendulums), and a last filter (Filter 7). Below this is the marionette, from which the test mass is suspended by 480 micron-fused silica fibres. To achieve vertical attenuation, the filter chain is equipped with triangular maraging blades and magnetic anti-springs.

The attenuation of the whole suspension from the ground is passive. Active controls are used to damp the structural modes (from 0.03–2 Hz).

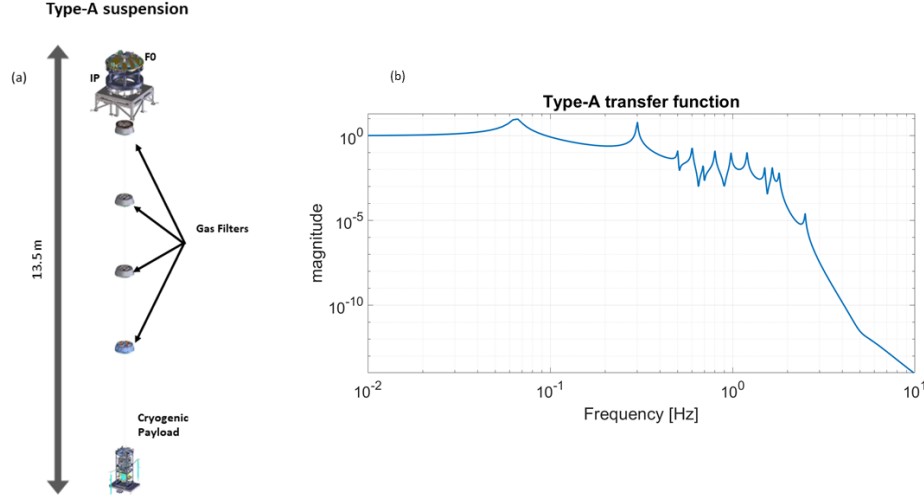

**Figure 8.** Panel (**a**) shows a technical drawing of a Type A system. From top to bottom, the isolation stages are: the pre-isolator (inverted pendulum and top filter—F0), four GAS filters, and the cryogenic payload (platform, intermediate-mass, marionette, and mirror) [23]. Panel (**b**) shows the Type A mechanical transfer function from the ground to the test mass along the longitudinal degree of freedom.

For completeness, Figure 10 shows the ground displacement, measured at the Cascina site, filtered by means of the SA and compared to AdV sensitivity curve. Looking at this Figure, it is clear that it is thanks to the high-performance suspension system that the current second-generation GW antennas have been able to improve their sensitivity down to 10 Hz. The plan for the future third-generation GW detectors, such as ET [26] and the Cosmic Explorer (CE) [27], is to extend the detection band below 10 Hz. With this intent in mind, an upgrade of the suspension systems is necessary in order to improve seismic attenuation in the low frequency band. Other technological improvements such

as cryogenic-payloads and a Newtonian noise subtraction system (see Section 4) will be necessary. Moreover, ET is intended to be sensitive down to 2–3 Hz; for this reason, it will be built underground (see Section 2.2).

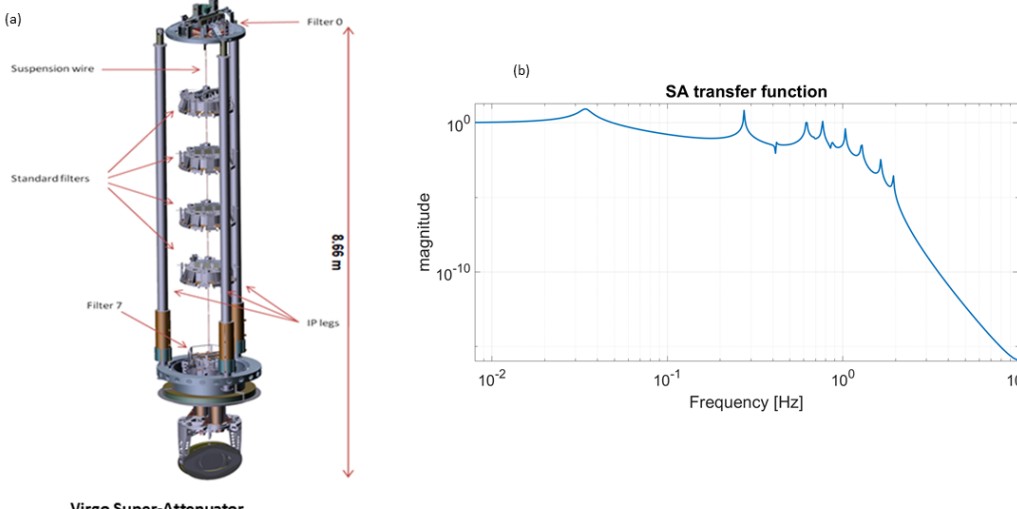

**Figure 9.** Panel (**a**) shows a technical drawing of the Virgo SA. From top to bottom, the isolation stages are: pre-isolator (inverted pendulum and top filter—F0), five filters and the payload (marionette and mirror). Panel (**b**) shows the SA mechanical transfer function from the ground to the test mass along the longitudinal degree of freedom.

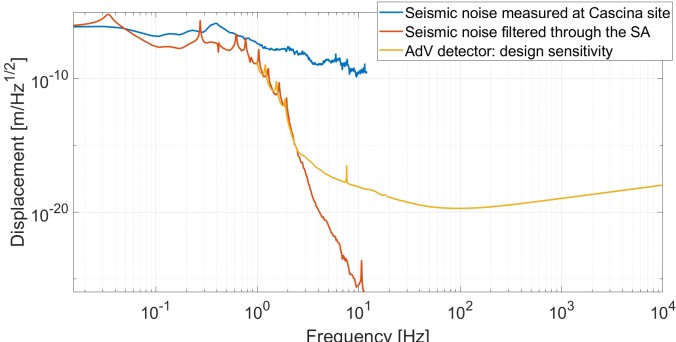

**Figure 10.** Ground displacement not filtered (blue) and filtered through the SA (red) compared to the design sensitivity of the AdV detector.

## 4. Newtonian Noise

NN has been foreseen since the beginning of the GW detector era [28], and in 1998, with the contemporaneous works of Beccaria et al. [29] and Hughes and Thorne [30], it was predicted to become the last sensitivity wall in the low frequency band (<30 Hz) of ground-based interferometric detectors. However, it is only with the advent of advanced [31] and third-generation [32] GW detectors that NN will start to threaten their sensitivity and to become the dominant source of noise in the low frequency band. Indeed, in the forthcoming observing runs, quantum and thermal noise will be lowered enough to allow NN to become the dominant noise. For this reason, it will be of utmost importance to reduce NN as much as possible.

Addressing the NN problem will also be a crucial task for 3rd generation gravitational-wave detectors, such as ET and CE.

Due to its gravitational nature, NN cannot be physically shielded or reduced without forcing severe modifications to the already existing infrastructure [30,33]. Therefore, the best approach to suppress it is through active noise cancellation. This aspect will be covered in more detail in Section 6.

The gravity fluctuations induced by a density variations are a simple consequence of the Newtonian law

$$\delta\phi(\vec{r}_0, t) = -G \int dV \frac{\delta\varrho(\vec{r}, t)}{|\vec{r}_0 - \vec{r}|} \tag{7}$$

where $\varrho(\vec{r}, t)$ represents a density variation at a position $\vec{r}$ and at a time $t$. $G$ is the gravitational constant. Every mechanism that produces a density variation in the media surrounding the test masses represents a source of Newtonian noise. Thus, we can have

$$\delta\varrho_{\text{seis}}(\mathbf{r}, t) = -\nabla \cdot (\rho_{\text{soil}}(\mathbf{r})\boldsymbol{\xi}(\mathbf{r}, t)) \tag{8}$$

$$\delta\varrho_{\text{press}}(\mathbf{r}, t) = \frac{\bar{\rho}_{\text{atm}}}{\gamma \bar{p}_{\text{atm}}} \delta p_{\text{atm}}(\mathbf{r}, t) \tag{9}$$

$$\delta\varrho_{\text{temp}}(\mathbf{r}, t) = -\frac{\bar{\rho}_{\text{atm}}}{\bar{T}_{\text{atm}}} \delta T_{\text{atm}}(\mathbf{r}, t) \tag{10}$$

where $\rho_{\text{soil}}(\mathbf{r}, t)$ is the density of the soil; $\boldsymbol{\xi}(\mathbf{r}, t)$ is the seismic displacement; $p_{\text{atm}}(\mathbf{r}, t)$ and $T_{\text{atm}}(\mathbf{r}, t)$ are the air pressure and temperature; $\gamma \sim 1.4$ is the adiabatic index; and $\bar{\rho}_{\text{atm}}$, $\bar{p}_{\text{atm}}$, and $\bar{T}_{\text{atm}}$ are the average density, pressure, and temperature of the atmosphere, respectively. In the next two Subsections, ANN and SNN will be presented in more detail.

### 4.1. Atmospheric Newtonian Noise

All the phenomena that can perturb the local gravity field on timescales less than 0.5 s are relevant to present and future ground-based GW detectors. Above this value, their sensitivity is too degraded for NN to be a concern. ANN can be produced by infrasound waves (pressure waves) and temperature fluctuations, as well as shock waves and turbulent phenomena.

Saulson [34] was the first to consider NN produced by atmospheric pressure fluctuations. Creighton [35] analyzed the topic, also discussing the production of NN due to advected temperature fields, high speed objects, and shock waves. Regarding pressure fluctuations generated by turbulent flow, the first model was made in [36] and then resumed in [2], in which a complete review of NN is given.

NN produced by infrasound pressure waves can be divided into two contributions: external and internal. The external contribution is produced by infrasound waves propagating in the atmosphere, which can even be produced by sources far from the detector; indeed, acoustic waves can propagate for long distances with negligible attenuation in the atmosphere [37]. Wind turbines are an example of infrasound and seismic noise sources that should be avoided in the proximity of GW detectors [38,39]. The internal contribution is instead due to the noise generated by the machinery hosted in the buildings (Virgo/LIGO) or caverns (KAGRA/ET). This noise must of course be avoided and mitigated as much as possible. Some studies have been conducted in order to assess its contribution inside underground caverns [40,41].

Concerning the NN generated by the advected temperature field, some more considerations are necessary. Pockets of warm air are mixed with pockets of cool air by convective turbulence, but given the timescales we are interested in, we can consider them as frozen in the atmosphere. However, they can produce gravitational noise when the air flow transports these pockets past the detector. Creighton [35] calculated the NN induced by the density fluctuations caused by the advection of these air pockets. The calculation must involve statistical considerations about the temperature field. Considering a uniform airflow parallel to the ground, we can see that the NN induced by temperature advection strongly depends on wind velocity (see Figure 11). In Figure 11 the ANN produced at the surface is compared with the ET-D sensitivity curve [42]. This shows why building the ET underground is important; reaching the targeted sensitivity would be very challenging in a surface location. Going underground would instead reduce the ANN to a level well below the ET sensitivity curve; indeed, the greater the distance from the source of noise, the lower the induced NN (see Table 1).

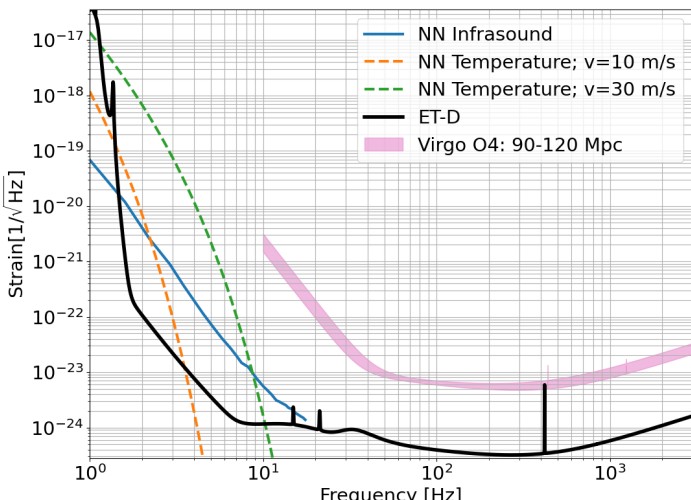

**Figure 11.** ANN generated by external infrasound and advected temperature fields at the Earth's surface compared with the sensitivity of Virgo foreseen for O4 and with the ET-D sensitivity curve. The ET curve was plotted only to show that building ET on the surface would lead to the requirement of reducing ANN generated by infrasound and advected temperature fields of some orders of magnitude. The ANN produced by infrasound and advected temperature fields were calculated with the models of [35]. The curves for the NN from advected temperature fields were estimated for wind velocities of 10 m/s and 30 m/s. The necessary data to plot the NN produced by infrasound fields were taken from the 75th percentile of Figure 3 in [43].

**Table 1.** Comparison between the square root of the power spectral densities of the ANN generated by different mechanisms: $h$ is the depth of the underground interferometer, $R$ is the radius of the cavern/room, and $d$ is the shortest distance between the test mass and the moving air flow.

| NN Type | Coupling Factor | Condition |
|---|---|---|
| External Infrasound | $\propto 1/h^2$ | $h \gg \frac{c_s}{4\pi f}$ |
| Internal Infrasound | $\propto R^2$ | $R \ll \frac{c_s}{2\pi f}$ |
| Advected Temperature | $\propto e^{-2\pi f d/v}$ | $d \gg \frac{v}{2\pi f}$ |

Table 1 shows how the square root of the power spectral density (PSD) of the strain generated by the different sources of ANN depends on some geometrical parameters such as the distance of the test mass from the atmosphere (external infrasound), the radius of a cavern/room (internal infrasound), and the the minimum distance between the test mass and the air flow (temperature advection). From here, we can easily see that an underground detector would greatly reduce the influence of ANN on the test masses. However, for shallow underground detectors, ANN might still need to be reduced. This can be accomplished by monitoring atmospheric pressure and temperature; however, monitoring the pressure is quite troublesome. Indeed, the infrasound microphones can be affected by nearby turbulent flows (also called wind noise) that can mask the signal in which we are interested [44,45]. This reduces the coherence between two microphones, thus worsening the capabilities of cancelling the NN (see Section 6).

The LIDAR system is a good candidate to measure pressure and temperature fields, but its resolution is not yet sufficient [46] for pressure measurements. Indeed, we are interested in phenomena happening on temporal scales smaller than T = 0.5 s. This means that for pressure waves we need a spatial resolution smaller than $Tc_s/2\pi \sim 27$ m, where $c_s$ is the sound speed in the air ($\simeq$340 m/s).

The possibility of exploiting cosmic rays showers to monitor atmospheric parameters such as the temperature or the pressure has been proposed as well [47]; however, this technique requires a sufficient spatial and temporal resolution (T < 0.5 s) to resolve the fast changes in the density which could give rise to ANN.

*4.2. Seismic Newtonian Noise*

SNN is produced by seismic waves that are able to locally modify the density of the medium that they are crossing. Rayleigh waves produce density variations through the characteristic elliptical movement that they induce on particles, whereas P-waves occur through the compression and decompression of the soil (see Section 2). Concerning S-waves, we need to be more careful. Shear waves do not produce any density fluctuation (and thus SNN) in a homogeneous volume, but they can if discontinuities (such as a cavern) are present. This is an important point for underground detectors, which are hosted in caverns. The SNN produced by the soil surrounding a cavern of radius $a$ can be written as [2]:

$$\delta \mathbf{a}^{\text{body-waves}}(\mathbf{0},t) = 4\pi G \rho_0 \left( 2\boldsymbol{\xi}^P(\mathbf{0},t) \frac{j_1(k^P a)}{k^P a} - \boldsymbol{\xi}^S(\mathbf{0},t) \frac{j_1(k^S a)}{k^S a} \right) \tag{11}$$

where $\delta \mathbf{a}^{\text{body-waves}}(\mathbf{0},t)$ is the acceleration induced by the SNN at the center of the cavern; $G$ is the gravitational constant; $\boldsymbol{\xi}^P$ and $\boldsymbol{\xi}^S$ are the displacement induced by P- and S-waves, respectively; $j_i(x)$ is the spherical Bessel function of order $i$; and $k^P$ and $k^S$ are the absolute value of the wave vector of P and S waves, respectively. The equation is obtained by considering a homogeneous and isotropic space with density $\rho_0$ and the presence of a cavity of radius $a$. We can note that P-waves contribute to $\delta \mathbf{a}$ with a factor of 2 compared to S-waves. This happens because P-waves contribute to the NN by generating density variations in two ways: through the compression/decompression of the ground and through the displacement of cavity walls. S-waves, instead, enter into play by only generating density variations at the cavity wall discontinuity.

Scattering of seismic waves in the cavern could cause the conversion of P-waves into S-waves and vice versa. However, in [2] it is shown that in the limit of long wavelengths ($k^{P,S}a \to 0$), one can neglect this scattering.

Rayleigh waves dominate the vertical seismic spectrum at the surface and for this reason they are the main source of SNN in detectors such as Virgo and LIGO. These waves are exponentially attenuated underground and contribute less to underground SNN, which instead is dominated by body waves (see Equation (11)). We can model the SNN produced by Rayleigh waves for a surface detector with test masses suspended at $h$ m from the ground as follows [2]:

$$\delta \mathbf{a}^{\text{Rayleigh}}(\mathbf{r}_0,t) = 2\pi G \rho_0 \gamma(\nu) e^{-hk_\rho} \xi_z(\mathbf{0},0) e^{i(\mathbf{k}_\rho \cdot \mathbf{r}_0 - \omega t)} \begin{pmatrix} i\cos(\theta) \\ i\sin(\theta) \\ -1 \end{pmatrix} \tag{12}$$

where $\gamma(\nu)$ is a factor determined by the Poisson ratio (i.e. by the elastic properties of the medium) and its value ranges from 0.5 to 1, and $\theta$ is the angle that the horizontal wave vector $\mathbf{k}_\rho$ forms with the $x$-axis. We obtain Equation (12) by calculating the gravity potential induced by a Rayleigh wave, taking its gradient with respect to $\mathbf{r}_0$ and solving for a test mass located above the surface.

However, this model neglects the topology. Indeed, the scattering of seismic waves from an irregular surface topography can vary the composition of the seismic field, leading to complex structures that are not completely characterized by surface displacement [48]. Moreover, for a complete SNN estimation of surface detectors, the underground body wave contribution should also be added to the Rayleigh wave surface contribution.

Rayleigh waves propagating below the surface will generate SNN in an underground test mass through three main mechanisms: surface displacement, rock compression/decompression, and cavern wall displacement. We need to coherently add all these effects to obtain their SNN contribution in the test mass. The PSD of the NN strain induced by Rayleigh waves in an underground detector is given in Equation (2) of [37]. In Figure 12, seismic noise and NN of seismic origin are compared with the ET-D sensitivity curve. The estimate for the NN from Rayleigh waves is calculated using a seismic spectrum lying exactly in the middle between

the Peterson NLNM and NHNM (see Figure 2). The estimates for the seismic noise and the SNN from body waves are instead calculated using five times the NLNM. The seismic noise curve was obtained by filtering the ground motion with the transfer function of the 17 m tall suspensions envisaged for ET [42]. The SNN from body waves was calculated based on Equation (11), with a negligible cavity radius compared to the wavelength of the incoming seismic waves and considering an equal energy partition between the two polarizations of S-waves and P-waves. Finally, the SNN produced by Rayleigh waves was estimated using the model of Equation (2) in [37]. However, this estimate must be taken with a grain of salt; indeed, it strongly depends on the velocity dispersion model of Rayleigh waves, as well as on the velocities of P and S-waves.

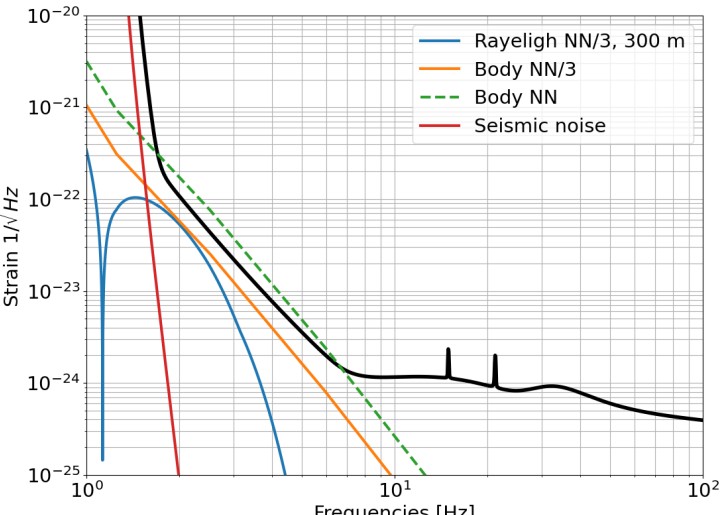

**Figure 12.** ET-D sensitivity curve compared with the seismic noise and NN of seismic origin. The seismic noise curve was obtained using a seismic spectrum lying exactly at the midpoint between Peterson's NLNM and NHNM (see Figure 2). The same spectrum was used to evaluate the NN generated by Rayleigh waves. For the NN from body waves, we used 5 times the Low Noise Model spectrum, as was done in [37].

## 5. Other Sources of Newtonian Noise

In principle, NN can be generated by any mass displacement. This means that water flow in underground caverns can also produce such noise. In KAGRA, there can be a large amount of water flowing in the pipes during spring (approximately 1200 tons of water at most per year passes through the drainage pipes near the Y-end station [23]). This could lead to sensitivity limitations of the detector. Indeed, although NN produced by water compression can be considered negligible, the water surface profile can vary during its flow, thus leading to a density variation (the same mechanism by which S-waves can produce NN in presence of a discontinuity). Preliminary models [49] show that NN from turbulent water flow could limit KAGRA's sensitivity.

Another possible source of concern is the NN generated by the vibrations of the cryogenic shielding or by boiling cryogenic liquids. E. Bonilla et al. in [50] studied these effects in relation to the development of LIGO Voyager [51]. They concluded that these sources should not constitute a concern since their NN production will lie below the sensitivity curve of Voyager, as well as that of ET. Structure vibrations will matter only if they have very high Q-factors that can amplify vibrations. Indeed, in [2] (Section 6.5) J. Harms shows that the interaction between an oscillating point mass close to the GW test mass is proportional to the oscillation amplitude, but it is also suppressed by the relative distance between the two objects. This means that, in order to threaten the detector sensitivity, the amplitude should be high and the oscillating objects should be very close to the test mass.

### 6. Newtonian Noise Cancellation

Given that NN is generated by fluctuations in the gravity field, we cannot easily shield the detector from NN fluctuations. Excavating meter-scale recesses around the test masses would help in reducing nearby density variations and the consequently induced NN by a factor 2–4: this was shown in [33]. A. Singha et al. evaluated, through numerical simulations, the NN originating from an isotropic Rayleigh field in Virgo with and without the recess structure hosting the test masses' clean rooms. They showed that the presence of this empty space leads to a reduction of a factor of two in the estimated NN of Virgo. This result is also supported by the findings of [52], in which seismic data collected in the Virgo West End Building were used to make a first estimate of the SNN affecting the test mass. Concerning ET, cavern dimensions should ideally help in reducing SNN. Plotting the P and S contributions of Equation (11) as a function of $k^P a$ shows that if the cavity radius is 0.4 times the seismic wavelength, then gravity perturbations are reduced by a factor of two. Translated in terms of cavern dimensions, we see that at 30 Hz (the highest interesting frequency for the NN) and considering a conservative P-wave velocity value of 4 km/s, the minimal radius to reduce NN should be of about 50 m [2]. It is clear that such a large radius is unfeasible for underground caverns. In fact, the high rock stress could lead to instabilities that might be fatal [53]. For this reason, it is important to think of other more effective ways to reduce NN both in underground and surface detectors.

Another possible way to reduce both SNN and seismic noise would be the employment of seismic metamaterials, which should provide seismic cloaking that is able to reduce the incoming seismic waves [54,55]. This approach, like the previous one, entails major modifications of the detector infrastructure, so it is not easy to implement in existing GW detectors. Indeed, this would imply building periodic structures around the test masses with dimensions of the order of the seismic waves of interest.

The most immediate way to reduce NN in GW detectors is through active noise cancellation. This method was first suggested by Hughes and Thorne in [30] and then developed by G. Cella in [56]. Active noise cancellation has already been employed in GW interferometers to further suppress noise in the instrument [19–21]. In LIGO, for example, this method is used to actively suppress seismic noise [22].

The basic idea underlying active noise cancellation is to monitor a noise source with some witness sensors and then use these data as an input to a linear filter to reconstruct the noise. In the case of NN, this is possible because the seismic displacement and the induced NN have a linear relationship (see Equations (7) and (8)). The optimal linear filter is calculated to minimize the square error between the estimated signal (the NN) and the real signal and it is known as a Wiener filter (WF) [57].

The main issue in NN cancellation is represented by finding the optimal positions of the N witness sensors in order to maximize the WF capabilities to estimate the SNN. This can be achieved in the frequency domain, but for the final subtraction of NN from GW data, the WF will have to be implemented in the time domain. The distinction between the time-domain WF and the frequency-domain WF lies principally in the fact that in the frequency domain the information regarding the order of the filter disappears. The WF in the frequency domain can be expressed as

$$\hat{X}(\omega) = \mathbf{W}^T(\omega)\mathbf{Y}(\omega) \tag{13}$$

where both $\mathbf{W}(\omega)$ and $\mathbf{Y}(\omega)$ are N-dimensional vectors containing the Fourier transforms of the WF coefficients and the N witness signals, respectively. The WF is then defined by the coefficients that minimize the ensemble average of the square error function, $E[e^*[\omega]e[\omega]]$, with $e[\omega] = X[\omega] - \hat{X}[\omega]$:

$$\mathbf{W} = (\bar{\mathbf{P}}_{YY})^{-1}\mathbf{P}_{XY} \tag{14}$$

where $P_{YY_{ij}}(\omega) = E[Y_i^*(\omega)Y_j(\omega)]$ is the element ij of the $N \times N$ matrix of the Cross-PSD$_s$ between the $i^{th}$ and $j^{th}$ witness sensors and it is denoted as $\bar{\mathbf{P}}_{YY}$. $P_{XY_i}(\omega) = E[Y_i^*(\omega)X(\omega)]$ is the $i^{th}$ element of the the $N$-vector of the Cross-PSD between the target signal (the NN in

the detector) and the i$^{th}$ witness sensor signal; it is denoted as $\mathbf{P}_{XY}$. With this result we can define the residual as $R = E[e^*e]/P_{XX}$ and obtain

$$R(\omega) = 1 - \frac{\mathbf{P}_{XY}^\dagger \bar{\mathbf{P}}_{YY}^{-1} \mathbf{P}_{XY}}{P_{XX}} \tag{15}$$

where $P_{XX}(\omega) = E[X^*(\omega)X(\omega)]$ is the PSD of the target signal. Note that $P_{XX}$ is a scalar and it is real ($P_{XX} = P_{XX}^*$), whereas $\mathbf{P}_{XY}$ and $\bar{\mathbf{P}}_{YY}$ are a complex vector and a complex matrix, respectively.

Equation (15) indicates an important point: for effective NN cancellation, correlations between sensors and between them and the test mass are of the utmost importance, but until NN is dominated by other noises (in particular thermal and quantum noises, but also technical noises), evaluating the correlations between the sensors and the test mass strain is hopeless. For this reason, in order to optimize the seismic array positions, we need to rely on coupling models between the seismic displacement and the test mass (see Equation (4) in [52]). The optimal array should be in a configuration that maximizes the correlations between sensors (the elements of $\bar{\mathbf{P}}_{YY}$). We will come back to this point in the following Subsection, regarding underground NN cancellation.

Ideally, to perform NN cancellation, it would be sufficient to evaluate the WF coefficients of Equation (14). However, this should be done in the time domain, thus implying the inversion of a huge matrix (of the order of NPxNP, since we have to consider also the filter order, P). This could lead to statistical and numerical errors. A way around this would be using a gradient descent algorithm to find the WF coefficients, instead of calculating them through a matrix inversion [58].

In order to perform the cancellation, other sensors rather than seismometers have been proposed. Concerning surface detectors, we know that they are dominated by Rayleigh waves, which can be monitored using an array of vertical seismic sensors. However, a single tiltmeter located under the test mass would be enough to cancel the NN from surface detectors with the only limitation being its own self-noise [59]. The reason that a tiltmeter could perform better is that the vertical displacement under a test mass has zero correlation with the induced NN, whereas the horizontal one correlates in the direction of the test mass displacement. Using seismic sensors to record the horizontal channel would not help, since they are affected by the presence of Love waves, which would spoil the correlations. The tiltmeter, on the other hand, does not have this problem and it could be used for NN cancellation purposes. The possibility of using a tiltmeter is yet to be investigated for the underground case, where instead of Rayleigh waves, there are P and S-waves which mix (see next Subsection).

Another possibility for performing NN cancellation is represented by deep neural networks. Some preliminary works have already been carried out to check how they could perform compared to the WF [60] and to reconstruct the underground seismic field [61].

*Newtonian Noise Cancellation in Surface and Underground Detectors*

They key to good NN cancellation relies in the search for the optimal array, in order to maximize the WF capabilities to estimate the NN affecting the test mass. Finding such an array means searching for the array configuration that minimizes the residual of Equation (15) for a fixed frequency and number of sensors. If the seismic field is isotropic and homogeneous, this can be accomplished quite easily, especially for surface detectors. In reality, the seismic field, especially at the surface, is never isotropic and/or homogeneous. This leads to the need to use seismic data to reconstruct the surface seismic field by building a surrogate model that will be used to find the optimal array configuration [52]. This has been achieved for Virgo, leading to the array configuration shown in Figure 13 (left).

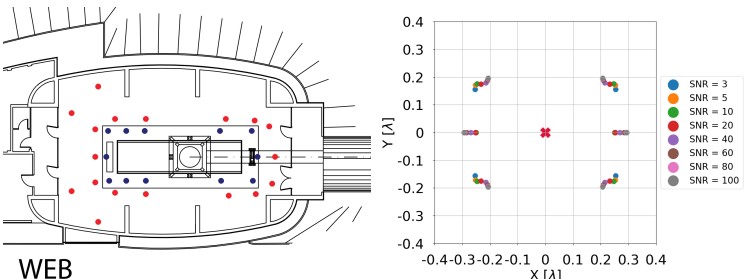

**Figure 13.** Comparison between the optimal array obtained for the West End Building (WEB) in Virgo (**left**) and the optimal array obtained for an isotropic and homogeneous Rayleigh wave field (**right**). The blue dots in the **left** figure represent sensors placed on the tower platform (which is anchored to the bedrock with pillars), whereas the red dots represent sensors placed on the floor of the building. The optimal array in the **right** figure is represented with different values of SNR. The *right* figure was taken from [62].

In underground detectors, correlations between seismic sensors are greatly spoiled by the mixing of P- and S-waves. Indeed, seismic sensors are only sensitive to the displacement and not to the wave polarization. This means that both P- and S-waves will contribute to the recorded displacement and, since P- and S-waves are uncorrelated, the correlations between the sensors will be degraded. Figure 14 shows a comparison of seismic correlations between the origin and all the other points in the presence of a homogeneous and isotropic seismic field composed only of P-waves in one case (left) and of a mix of P- and S-waves in the other case (right).

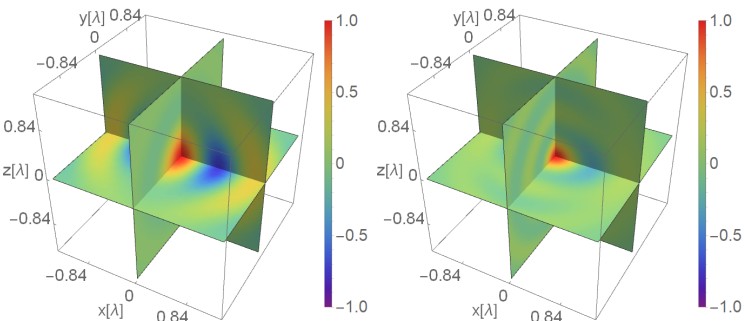

**Figure 14.** **Left**: seismic correlations between the origin and all the other points in the presence of a homogeneous and isotropic seismic field composed only of P-waves. **Right**: seismic correlations between the origin and all the other points in the presence of a homogeneous and isotropic seismic field composed of 1/3 of P-waves and 2/3 of S-waves (assuming an equal distribution of energy between the three polarizations: one for P-waves and two for S-waves). Figure taken from [62].

For this reason, the WF in the underground environment will never reach the limit imposed by the self noise of the sensors since it is already limited by degraded correlations. This is perhaps the biggest conceptual problem related to NN subtraction in underground detectors because it implies the need for a higher number of sensors to reach a given NN reduction level. Finding a way to disentangle P- and S-components would lead to better performance by the WF (increasing the correlations that could be evaluated separately). Employing tiltmeters, for example, could help in this, but this is something that remains to be investigated.

## 7. Conclusions

In this paper, two forms of noise present in interferometric GW detectors were reviewed: seismic and Newtonian noise. Newtonian noise has a seismic noise component; it follows then that these two forms of noise are intimately connected and, for this reason, it is natural to address them in the same review.

Seismic noise and its origin are described in detail and an explanation of how this noise can affect GW detection and the techniques employed to reduce its effects in current GW detectors have been provided. Newtonian noise has been described as well: all its contributors (atmospheric, seismic, and some other contributors) were described together, along with the main work carried out in the field.

To conclude, we must point out that third-generation GW detectors require particular care in regard to the design of the suspension system, as well as the design of the Newtonian noise cancellation system. Indeed, seismic and Newtonian noise can become a limiting factor for the detector sensitivity in the low frequency band.

**Author Contributions:** L.T. and F.B. contributed equally. All authors have read and agreed to the published version of the manuscript.

**Funding:** This research received no external funding.

**Institutional Review Board Statement:** Not applicable.

**Informed Consent Statement:** Not applicable.

**Data Availability Statement:** Data needed for some figures were taken from Virgo seismic channels, from [43,63] and from ORFEUS, the European Infrastructure for seismic waveform data in EPOS [64].

**Acknowledgments:** The authors would like to thank Jan Harms, Luciano Di Fiore, and Joris van Heijningen whose insightful comments helped to improve the paper. They would also like to thank Rosario De Rosa, who kindly provided the seismic data of Sos-Enattos site shown in Figures 4 and 5, Kazuhiro Yamamoto, who kindly provided the seismic data to reproduce Figure 3, The Kagra collaboration and Ayaka Shoda who kindly provided the technical drawing of the Type A system, shown in Figure 2 of [23] and the data used in Figure 8.

**Conflicts of Interest:** The authors declare no conflict of interest.

## Abbreviations

The following abbreviations are used in this manuscript:

| | |
|---|---|
| AdV | Advanced Virgo |
| ANN | Atmospheric Newtonian Noise |
| BSC | Basic Symmetric Chamber |
| CE | Cosmic Explorer |
| ET | Einstein Telescope |
| NHNM | New High Noise Model |
| NLNM | New Low Noise Model |
| NN | Newtonian Noise |
| PSD | Power Spectral Density |
| SA | Super Attenuator |
| SNN | Seismic Newtonian Noise |
| WF | Wiener Filter |

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
