# Peer review of "Seismic and Newtonian Noise in the GW Detectors"

_galaxies, doi:10.3390/galaxies10010020_

Round 1

Reviewer 1 Report

This manuscript tries to give an overview on the seismic and the Newtonian noises in the Gravitational Waves detectors, and provides strategies adopted to mitigate them. The following are my comments and critique:

General:

  1. The manuscript needs to be edited for grammar and wording.
  2. Introduction: The authors give a short introduction of two types of noises. The readers will expect more information about other possbile noise, and how they will affect the GW detection. 
  3. In line 111-112, the author introduces another vibration source named as anthropogenic noise, but without any explanation. A few examples of this kind of source will be appreciated.
  4. The authors include many figures to show the results, however, the elaboration of them is not clear. Especially for figure 3 and 7c, there is no word mentioning it in the text. 

Author Response

Dear Referee, 

thank you for your valuable comments. In particular, we went through the grammar and the wording in a deeper way. We hope that now the paper is more readable and improved also from that point of view. 

 Moreover, we addressed all your comments in the text:

The manuscript needs to be edited for grammar and wording.

--> Done.

Introduction: The authors give a short introduction of two types of noises. The readers will expect more information about other possbile noise, and how they will affect the GW detection.

--> Done.

In line 111-112, the author introduces another vibration source named as anthropogenic noise, but without any explanation. A few examples of this kind of source will be appreciated.

--> Done.

The authors include many figures to show the results, however, the elaboration of them is not clear. Especially for figure 3 and 7c, there is no word mentioning it in the text. 

--> Done.

Thank you for your effort,

The Authors

Reviewer 2 Report

This article is a good review on the state of the art in seismic noise and Newtonian noise in gravitational-wave detectors. The authors discuss the seismic and atmospheric sources that produce these noises, the suspension/isolation systems employed to filter seismic noise, relevant computations for Newtonian noise, and Newtonian noise cancellation schemes for surface and underground detectors.
I only have minor comments on the content.

L63: “amplitude decreases exponentially with the distance from the source”: Perhaps I misunderstood the geometry in mind, but I assume what is meant here is that the amplitude of the wave decreases exponentially as a function of depth from the surface.

L65: “amplitudes of several cm”: This may be a typical value for earthquakes, but I would think the relevant value for GW detectors is the typical ambient fluctuation, which is more like micrometers.

Eq 1: Here x(f) denotes a spectral density, but later x(w) and similar expressions denote a complex frequency series. The authors may want to make the notation distinct.

L131: stray “Hz” for the Sos Enattos and Kamioka values

Eqs 4, 5, 6: These are all equations for undamped oscillators, but LL157-159 (and perhaps Fig 6) indicate the effect of a finite Q factor. I suggest including the effect of a finite Q in the equations, since it is a key experimental parameter for the suspension design.

L247: The use of “turbulent phenomena” here is a bit imprecise. Cafaro et al (and the related discussion in Harms) are concerned specifically with the Lighthill effect sourced by a fully developed turbulence in the open atmosphere. Other kinds of turbulent phenomena are potentially relevant (e.g., vortex generation near buildings, as in Creighton’s paper).

Fig 11 caption: I suggest “ANN at the Earth’s surface…” or similar just to make it clear to the reader that the underground suppression factors have not been applied to the NN curves.

L337: “5 times the NLNM”: From ref 33 this appears to be 5 times in amplitude; on the other hand, the reader is referred to Fig 2 which plots the power. The authors should make it clear that this is an amplitude factor (this also applies to some other factors mentioned, like on L372).

L359: To give the reader context, the authors should consider adding a reference to LIGO Voyager (doi:10.1088/1361-6382/ab9143)

Ref 3: information is missing

Author Response

Dear Referee, 

thank you for your valuable comments, we think that they really helped to improve our paper. 
Below you can find our answers to your comments. 

L63: “amplitude decreases exponentially with the distance from the source”: Perhaps I misunderstood the geometry in mind, but I assume what is meant here is that the amplitude of the wave decreases exponentially as a function of depth from the surface.

--> Yes, it was not written well. We expressed it in a better way.

L65: “amplitudes of several cm”: This may be a typical value for earthquakes, but I would think the relevant value for GW detectors is the typical ambient fluctuation, which is more like micrometers.

--> Right, we added that it is referred to Earthquakes. 

Eq 1: Here x(f) denotes a spectral density, but later x(w) and similar expressions denote a complex frequency series. The authors may want to make the notation distinct.

--> We used another notation for the spectral density. 

L131: stray “Hz” for the Sos Enattos and Kamioka values

--> Fixed

Eqs 4, 5, 6: These are all equations for undamped oscillators, but LL157-159 (and perhaps Fig 6) indicate the effect of a finite Q factor. I suggest including the effect of a finite Q in the equations, since it is a key experimental parameter for the suspension design.

--> Done.

L247: The use of “turbulent phenomena” here is a bit imprecise. Cafaro et al (and the related discussion in Harms) are concerned specifically with the Lighthill effect sourced by a fully developed turbulence in the open atmosphere. Other kinds of turbulent phenomena are potentially relevant (e.g., vortex generation near buildings, as in Creighton’s paper).

--> We changed the sentence in: "For what concerns pressure fluctuations generated by turbulent flow, a first model was made in (Cafaro et al.) and then [...]"

Fig 11 caption: I suggest “ANN at the Earth’s surface…” or similar just to make it clear to the reader that the underground suppression factors have not been applied to the NN curves.

--> Done

L337: “5 times the NLNM”: From ref 33 this appears to be 5 times in amplitude; on the other hand, the reader is referred to Fig 2 which plots the power. The authors should make it clear that this is an amplitude factor (this also applies to some other factors mentioned, like on L372).

--> we substituted fig 2 and now it shows the amplitude. 

L359: To give the reader context, the authors should consider adding a reference to LIGO Voyager (doi:10.1088/1361-6382/ab9143)

--> Done

Ref 3: information is missing

--> Fixed

Thank you for your effort,

The Authors